# Is information provided within chronic subdural haematoma education resources adequate? A scoping review

Conor S. Gillespie[1,2], Samuel Khanna[3], Mark E. Vivian[3], Samuel McKoy[2], Alvaro Yanez Touzet[4], Ellie Edlmann[1], Daniel J. Stubbs[3], Benjamin M. Davies[1]*

1 Department of Neurosurgery, Department of Clinical Neurosciences, University of Cambridge, Cambridge, United Kingdom, 2 The Walton Centre NHS Foundation Trust, Liverpool, United Kingdom, 3 University Division of Anaesthesia, University of Cambridge, Cambridge, United Kingdom, 4 School of Medical Sciences, Faculty of Biology, Medicine and Health, University of Manchester, Manchester, United Kingdom

* bd375@cam.ac.uk

**Data Availability Statement:** All relevant data are within the paper and its Supporting Information files.

## Abstract

### Background

Chronic subdural haematoma (CSDH) is becoming increasingly prevalent, due to an aging population with increasing risk factors. Due to its variable disease course and high morbidity, patient centred care and shared decision making are essential. However, its occurrence in frail populations, remote from specialist neurosurgeons who currently triage treatment decisions, challenges this. Education is an important component of enabling shared decisions. This should be targeted to avoid information overload. However, it is unknown what this should be.

### Objectives

Our objectives were to conduct analysis of the content of existing CSDH educational materials, to inform the development of patient and relative educational resources to facilitate shared decision making.

### Methods

A literature search was conducted (July 2021) of MEDLINE, Embase and grey literature, for all self-specified resources on CSDH education, and narrative reviews. Resources were classified into a hierarchical framework using inductive thematic analysis into 8 core domains: Aetiology, epidemiology and pathophysiology; natural history and risk factors; symptoms; diagnosis; surgical management; nonsurgical management; complications and recurrence; and outcomes. Domain provision was summarised using descriptive statistics and Chi-squared tests.

### Results

56 information resources were identified. 30 (54%) were resources designed for healthcare professionals (HCPs), and 26 (46%) were patient-orientated resources. 45 (80%) were

**Funding:** The authors received no specific funding for this work.

**Competing interests:** The authors have declared that no competing interests exist.

specific to CSDH, 11 (20%) covered head injury, and 10 (18%) referenced both acute and chronic SDH. Of 8 core domains, the most reported were aetiology, epidemiology and pathophysiology (80%, n = 45) and surgical management (77%, n = 43). Patient orientated resources were more likely to provide information on symptoms (73% vs 13%, p<0.001); and diagnosis (62% vs 10%, p<0.001) when compared to HCP resources. Healthcare professional orientated resources were more likely to provide information on nonsurgical management (63% vs 35%, p = 0.032), and complications/recurrence (83% vs 42%, p = 0.001).

## Conclusion

The content of educational resources is varied, even amongst those intended for the same audience. These discrepancies indicate an uncertain educational need, that will need to be resolved in order to better support effective shared decision making. The taxonomy created can inform future qualitative studies.

## Introduction

Chronic subdural haematoma (CSDH) is commonly encountered in neurosurgery [1]. With an incidence of 1.7–20.6 per 100,000 per year [2], cases are predicted to rise in line with an ageing population, increasing frequency of head injuries, and use of antiplatelet and/or anticoagulant medication [3]. This is forecast to increase operative workload by 50% in the next 20 years [4, 5].

The CSDH disease course can vary considerably, and several care decisions may be encountered. The principal decision is often whether surgical drainage is required, and how urgently. Surgery is offered to patients with symptoms resulting from mass effect such as neurological deficits and/or reduced consciousness level [1]. Further decisions may then include the type of surgery and anaesthesia, or increasingly the role of interventional radiology, although this is still at the experimental stage [6]. They may include the timing of surveillance imaging, or preoperative optimisation for patients on anti-thrombotics or with acute medical problems [7]. These decisions are often not clear cut, particularly given the co-existent morbidity and frailty amongst people with CSDH and must be individualised. To do this, decisions must involve the patients and their families/carers; the critical stakeholders [8].

For CSDH this process is often more challenging as neurosurgery is a tertiary speciality, and many hospitals will not have on-site access to neurosurgical services or specialists: In some centres, over 90% of cases may be referred from other non-specialist hospitals [9–11]. High-quality evidence in the care of CSDH is centred on operative and tertiary centre care–with randomised trials only providing clear evidence on some aspects of operative technique [12] and adjunctive steroid use [13]. This compounds the difficulties as care decisions pertinent to local hospitals are more likely to contain significant uncertainties, and instead rely on contextual information and judgement.

However before a shared decision making process can commence, patients and their families must acquire sufficient knowledge of the condition [14]. Healthcare professionals often employ aids to help bridge this knowledge gap. These could include websites, educational videos, and/or condition leaflets. Whilst potentially effective, designed by professionals these are at risk of imposing a bias on what information is portrayed, and/or can be generic in nature–poorly tailored to the decision/context in question. This can risk information overload, and potentially confuse and/or hinder an effective shared decision-making process [9]. In CSDH,

the content, appropriateness and availability of resources available to patients is poorly researched, with little information available [15–17].

Our objectives are aligned with this. Firstly, we sought to identify and explore the information themes that are found in educational resources for CSDH (available to professionals, patients, families, and caregivers). Our secondary objectives were to identify current knowledge gaps in the content of resources aimed at patients to healthcare professionals, to help in the development of future materials to facilitate shared decision making [18]. The overall objective of the study was to identify the information provided in CSDH educational resources.

## Materials and methods

### Resource type and categorisation

In line with health seeking behaviour of both the general public and healthcare professionals [19] potential educational information resources were defined as published articles (narrative reviews published in a peer-reviewed journal), other healthcare professional orientated resources, general health education websites and patient information leaflets. All resources available covering CSDH that could be considered educational resources, were included. Narrative reviews were selected because narrative reviews typically aim to provide educational content, and systematic reviews focus on specific clinical questions [20, 21]. This methodology was formulated from a previously published scoping review on educational resources in degenerative cervical myelopathy- this confirmed most systematic reviews examined a clinical research question rather than provide educational content [22]. To consider an information bias (i.e. whereby CSDH relevant information was missing from lay content), we included resources intended for healthcare professionals as well as patients and/or relatives.

Dedicated searches were performed for each media type, for content exclusively addressing CSDH. Content covering both acute (ASDH) and chronic SDH that did not differentiate between the two diagnoses, was excluded (Table 1). The search strategy is summarised below.

### Search strategy

A search was developed and refined for each of the four information types (Fig 1). For website-based resources, a hierarchical search strategy was employed to identify CSDH educational content with sequential searching of identified websites using page navigation, site search facilities, and finally in-page searching for the term 'Chronic Subdural Haematoma'.

**Table 1. Overall inclusion and exclusion criteria for screening resources to identify those with educational CSDH content.**

| Inclusion Criteria | Exclusion Criteria |
|---|---|
| English language | Do not specify CSDH in resource OR only specified ASDH |
| Data source identifies as an educational resource/tool OR is a narrative review | |
| Chronic Subdural Haematoma OR Acute Subdural Haematoma Plus Chronic Subdural Haematoma OR Head Injury Plus Chronic Subdural Haematoma | |

These criteria were applied to screen resources from the 4 key resource types: scientific research articles, other healthcare professional orientated content, health education websites and patient information leaflets. The aim was to identify public-facing resources that contained educational content on CSDH. Specific inclusion and exclusion criteria were then adapted for each resource type.

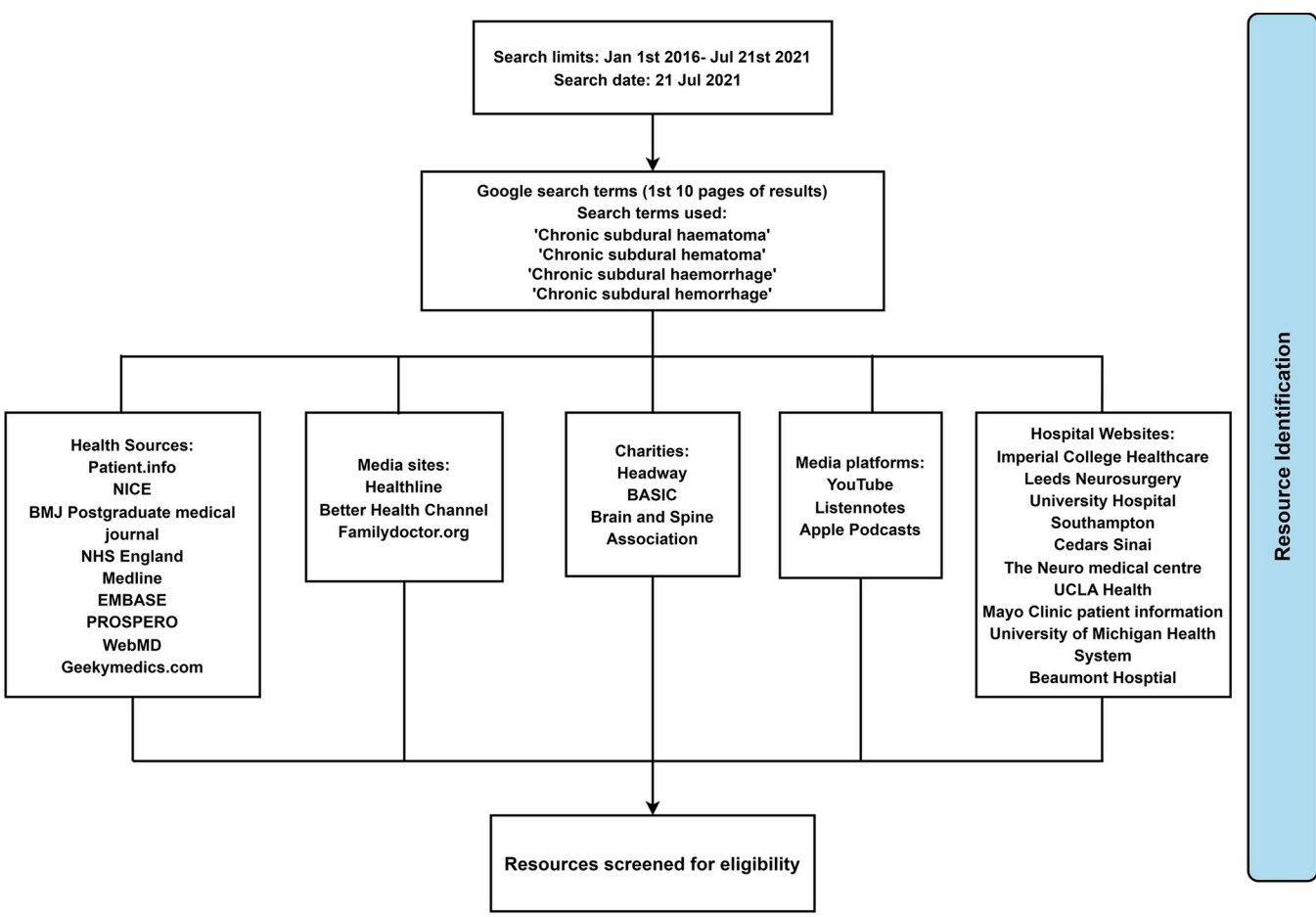

**Fig 1. Sources and searches used to identify resource.**

Websites were chosen based on their popularity as a health information resource, as defined in a previous review [22].

## Resource type acquisition and searching

For scientific literature, we used the term 'Chronic Subdural Haematoma' to search the EMBASE, Medline, and PROSPERO databases from 1st January 2016 until 21st July 2021 for narrative reviews. This time frame was selected to identify the most recent reviews published on CSDH, and to provide a representation of contemporary CSDH educational content. Videos and health education websites were identified by carrying out a search using Google (CA, USA) on 9th July 2021, including the top 10 results pages. For hospital patient information leaflets, we manually searched all United Kingdom (UK) and Republic of Ireland (ROI) Neurosurgical units listed on the Society of British Neurological Surgeons (SBNS) website [23], looking for CSDH educational resources. Searches were conducted by two authors (MEV and SK) with arbitration of inclusion and exclusion criteria by a third (DJS).

## Data extraction and coding

Educational information was extracted in duplicate by two independent authors (CSG and SM), and if disagreements could not be resolved by consensus, senior authors (DJS or BMD) were

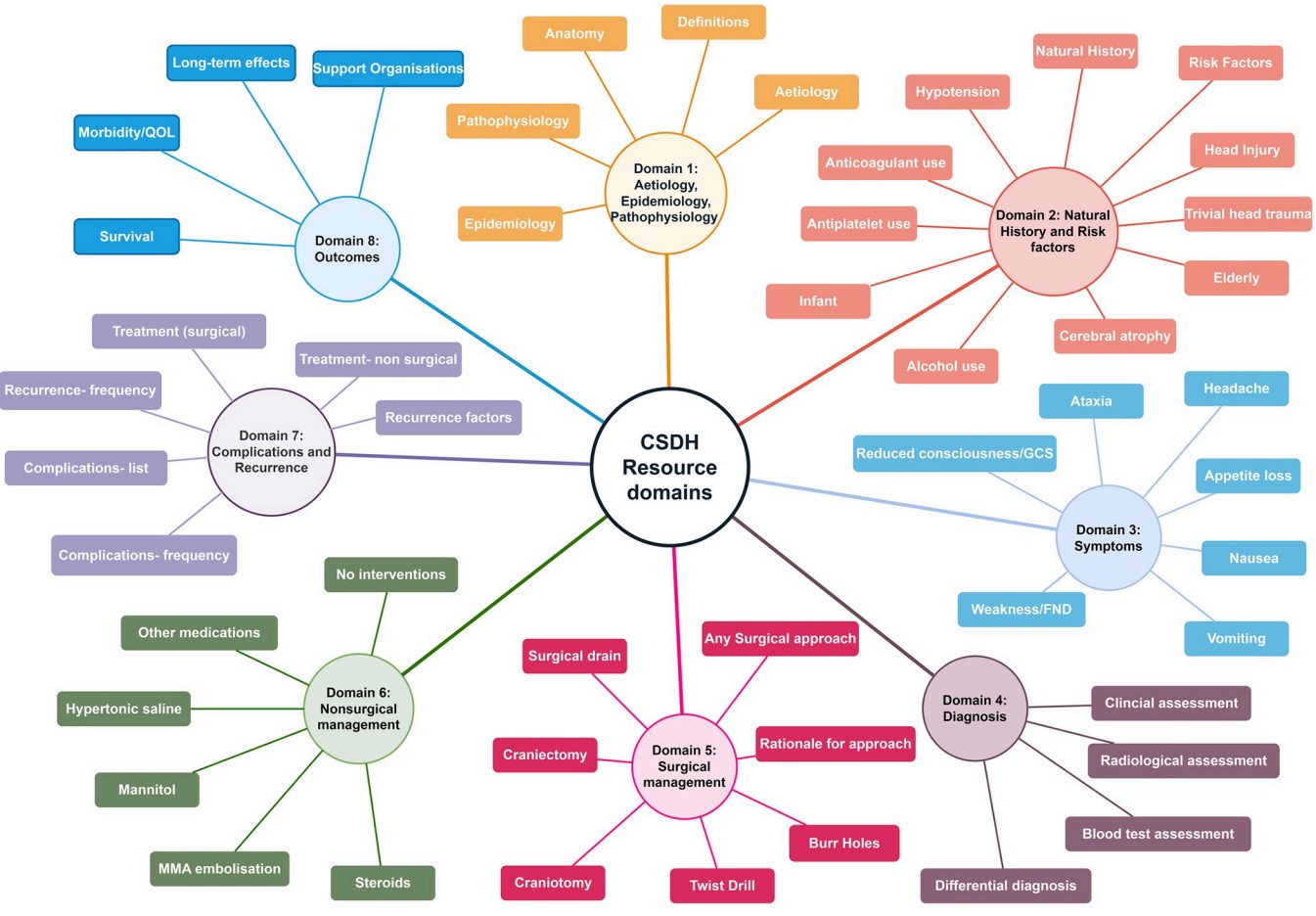

**Fig 2. Key education domains for CSDH, devised by inductive thematic analysis.**

consulted for clarification. Extracted information was coded by inductive thematic analysis [24] by two authors (CSG and BMD) into a hierarchical framework of 'domains' and 'subdomains'. This included eight domains (Fig 2): Aetiology, epidemiology and pathophysiology; natural history and risk factors; symptoms; diagnosis; surgical management; nonsurgical management; complications and recurrence; and outcomes, alongside a number of subdomains (Fig 2).

These were developed independently by assessing all information resources, and were refined through an iterative process until they were deemed applicable across all content, and consensus between authors was reached. For the domain 'symptoms', seven frequently encountered symptoms were selected by consensus to represent the subdomains [25]. Additional criteria were used to determine if a resource was targeted at a patient, carer or relative; healthcare professional (HCP) or both. A resource was defined as targeted at HCPs If it was published in a peer reviewed journal, or self-identified as 'being for use by healthcare professionals' specifically. All other resources were classified as targeted at patients. If a resource explicitly stated in its title or heading that it was for use by patients and HCPs, it was categorised as both.

## Statistical analysis

Data analysis was conducted using R V4.0.2, and figures created using RStudio (ggplot2, fsmb, and ggthemes packages). Data was summarised using descriptive statistics. The Chi square test

was used to compare differences between HCP resources and patient education resources except in the case of individual cell counts of five or less where Fisher's exact test was used. We considered a p value <0.05 to be significant.

## Results

Of the 100 potential resources identified from 10 results pages, 56 information resources were identified as eligible for inclusion. These included 26 patient-targeted resources, and 30 scientific articles targeted at HCPs. The 26 patient resources included 20 websites, 4 leaflets, 1 review article and 1 video. 45 (80%) were specific to CSDH, 11 (20%) represented head injury but included CSDH information, and 10 (18%) referenced both acute and chronic SDH.

### Core domains

The core domains for all resources are shown in Fig 3 and Table 2. The most common domains addressed were Aetiology, Epidemiology and Pathophysiology (80%, n = 45), Surgical management (77%, n = 43), and Natural history and risk factors (73%, n = 41). The least common included domains were Diagnosis (34%, n = 19), Symptoms (41%, n = 23) and Nonsurgical management (50%, n = 28). Most resources were designed for HCPs (54%, n = 30). Some were designed for patients, carers and the public (46%, n = 26), with 1 designed for both patients and HCPs (2%).

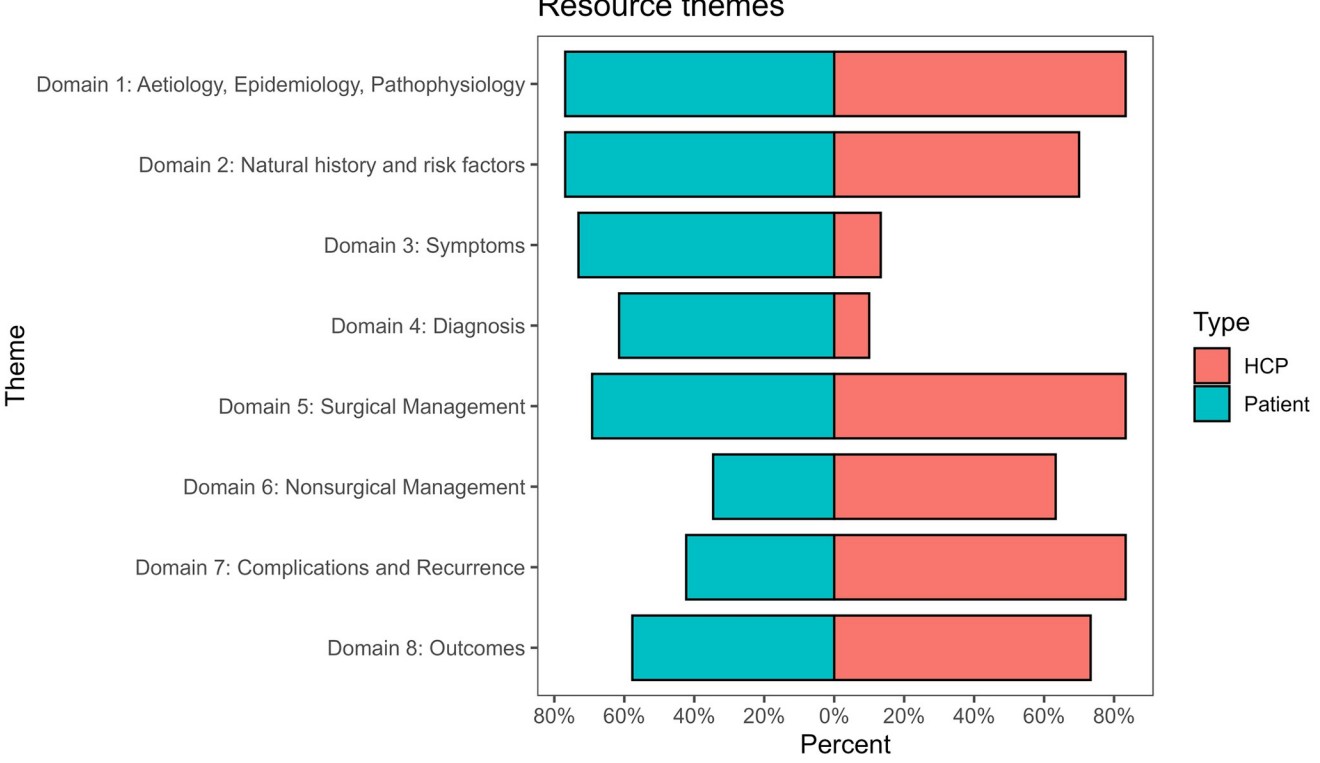

**Fig 3. Population pyramid of differences between domains, stratified by resource- target audience (Healthcare professional (HCP) or patient/relative-focused).**

**Table 2. Differences in domain components between patient orientated and healthcare professional orientated resources.**

| Title | Code | Overall (%, n) | Patient Resources %, (n) | Healthcare professional (HCP) resources % (n) | P value |
|---|---|---|---|---|---|
| **Aetiology, Epidemiology, Pathophysiology** | 1 | 80 (45) | 77 (20) | 83 (25) | 0.547 |
| Anatomy | 1a | 50 (28) | 69 (18) | 33 (10) | **0.007***|
| Definitions | 1b | 48 (27) | 65 (17) | 33 (10) | **0.017***|
| Aetiology | 1c | 55 (31) | 58 (15) | 53 (16) | 0.743 |
| Epidemiology | 1d | 38 (21) | 12 (3) | 60 (18) | **<0.001***|
| Pathophysiology | 1e | 46 (26) | 42 (11) | 50 (15) | 0.565 |
| **Natural history and risk factors** | 2 | 73 (41) | 77 (20) | 70 (21) | 0.560 |
| Natural history | 2a | 48 (27) | 58 (15) | 40 (12) | 0.186 |
| Risk factors | 2b | 67 (39) | 73 (19) | 67 (20) | 0.603 |
| Head Injury | 2bi | 50 (28) | 58 (15) | 43 (13) | 0.284 |
| Trivial head trauma | 2bii | 36 (20) | 39 (10) | 33 (10) | 0.690 |
| Elderly | 2biii | 57 (32) | 58 (15) | 68 (17) | 0.938 |
| Cerebral atrophy | 2biv | 41 (23) | 54 (14) | 30 (9) | 0.070 |
| Alcohol | 2bv | 32 (18) | 62 (16) | 7 (2) | **<0.001***|
| Infant | 2bvi | 5 (3) | 12 (3) | 0 (0) | 0.056 |
| Antiplatelets | 2bvii | 33 (18) | 52 (13) | 17 (5) | **0.005** |
| Anticoagulants | 2bviii | 45 (25) | 58 (15) | 33 (10) | 0.067 |
| Hypotension | 2bix | 5 (3) | 4 (1) | 7 (2) | 0.640 |
| **Symptoms** | 3 | 41 (23) | 73 (19) | 13 (4) | **<0.001***|
| Headache | 3a | 39 (22) | 69 (18) | 13 (4) | **<0.001***|
| Appetite loss | 3b | 5 (3) | 8 (2) | 3 (1) | 0.470 |
| Nausea | 3c | 30 (17) | 62 (16) | 3 (1) | **<0.001***|
| Vomiting | 3d | 30 (17) | 62 (16) | 3 (1) | **<0.001** |
| Weakness/Focal Neurological deficit | 3e | 36 (20) | 65 (17) | 10 (3) | **<0.001***|
| Reduced consciousness/GCS | 3f | 41 (23) | 73 (19) | 13 (4) | **<0.001***|
| Ataxia | 3g | 25 (14) | 46 (12) | 7 (2) | **<0.001***|
| **Diagnosis** | 4 | 34 (19) | 62 (16) | 10 (3) | **<0.001***|
| Clinical assessment | 4a | 27 (15) | 50 (13) | 7 (2) | **<0.001***|
| Radiological assessment | 4b | 36 (20) | 58 (15) | 17 (5) | **0.002***|
| Blood test assessment | 4c | 14 (8) | 23 (6) | 7 (2) | 0.085 |
| Differential diagnosis | 4d | 13 (7) | 27 (7) | 0 (0) | **0.003***|
| **Surgical management** | 5 | 77 (43) | 69 (18) | 83 (25) | 0.213 |
| Any Surgical approach | 5a | 70 (39) | 62 (16) | 77 (23) | 0.219 |
| Decision/rationale for surgical approach | 5b | 34 (19) | 39 (10) | 30 (9) | 0.505 |
| Burr Holes | 5c | 73 (41) | 62 (16) | 83 (25) | 0.066 |
| Twist Drill | 5d | 34 (19) | 8 (2) | 57 (17) | **<0.001***|
| Craniotomy | 5e | 59 (33) | 54 (14) | 63 (19) | 0.472 |
| Craniectomy | 5f | 7 (4) | 8 (2) | 7 (2) | 0.882 |
| Surgical drain | 5g | 25 (14) | 15 (4) | 33 (10) | 0.122 |
| **Nonsurgical management** | 6 | 50 (28) | 35 (9) | 63 (19) | **0.032***|
| No interventions | 6a | 32 (18) | 31 (8) | 33 (10) | 0.838 |
| Steroids | 6b | 27 (15) | 12 (3) | 40 (12) | **0.016***|
| MMA embolization | 6c | 20 (11) | 0 (0) | 37 (11) | **<0.001***|
| Mannitol | 6d | 4 (2) | 4 (1) | 3 (1) | 0.918 |
| Hypertonic saline | 6e | 2 (1) | 4 (1) | 0 (0) | 0.464 |
| Other medications | 6f | 29 (16) | 8 (2) | 47 (14) | **0.001***|
| **Complications and recurrence** | 7 | 64 (36) | 42 (11) | 83 (25) | **0.001***|

(*Continued*)

**Table 2.** (Continued)

| Title | Code | Overall (%, n) | Patient Resources %, (n) | Healthcare professional (HCP) resources % (n) | P value |
|---|---|---|---|---|---|
| Frequency of Recurrence | 7a | 46 (26) | 8 (2) | 80 (24) | **<0.001*** |
| Factors associated with recurrence | 7b | 46 (26) | 12 (3) | 77 (23) | **<0.001*** |
| Recurrence treatment- surgical | 7c | 25 (14) | 4 (1) | 43 (13) | **<0.001*** |
| Recurrence treatment- non surgical | 7d | 27 (15) | 12 (3) | 40 (12) | **0.016*** |
| Complications- frequency | 7e | 48 (27) | 31 (8) | 63 (19) | **0.015*** |
| Complications- list | 7f | 48 (27) | 31 (8) | 63 (19) | **0.015*** |
| **Outcomes** | 8 | 66 (37) | 58 (15) | 73 (22) | 0.218 |
| Survival | 8a | 36 (20) | 15 (4) | 53 (16) | **0.003*** |
| Morbidity/Quality of Life | 8b | 46 (26) | 54 (14) | 40 (12) | 0.300 |
| Long term effects | 8c | 50 (28) | 54 (14) | 47 (14) | 0.592 |
| Support organisations | 8d | 21 (12) | 46 (12) | 0 (0) | **<0.001*** |

## Differences between HCP resources and patient resources

The differences in information provided between the two groups are shown in Table 2 and Fig 3. There was a significant difference in reporting of four domains- patient resources were more likely to provide information on Symptoms (73% vs 13%, p<0.001) and Diagnosis (62% vs 10%, p<0.001). HCP resources were more likely to provide information on Nonsurgical management (63% vs 35%, p = 0.032), and complications and recurrence (83% vs 42%, p = 0.001).

## Domain summaries

**Domain 1: Aetiology, epidemiology, and pathophysiology.** Commonly reported sub-domains included Aetiology (55%, n = 31) and Anatomy (50%, n = 28). Patient resources more frequently defined CSDH, separating it from an ASDH (65% vs 33%, p = 0.017) and described anatomy (69% vs 33%, p = 0.007), whilst HCP resources were more likely to describe epidemiology (60% vs 12%, p<0.001).

**Domain 2: Natural history and risk factors.** The most commonly reported sub-domain was the reporting of any risk factors such as head injury, alcohol misuse, and medications (70%, n = 39). For specific risk factors reported, patient resources described more alcohol misuse (62% vs 7%, p<0.001), and antiplatelet medication use (52% vs 17%, p = 0.005) than HCP resources.

**Domain 3: Symptoms.** The most commonly reported symptoms overall were confusion/loss of consciousness (41%, n = 23), headache (39%, n = 22), and weakness/focal neurological deficit (36%, n = 20). Patient resources covered 6/7 symptoms more frequently than HCP resources (headache, appetite loss, nausea, vomiting, weakness/FND, and reduced consciousness level/GCS).

**Domain 4: Diagnosis.** The most commonly reported component of diagnosis was radiological assessment (36%. N = 20). HCP resources reported all sub-domains more frequently than patient resources.

**Domain 5: Surgical management.** Surgical management was highly reported in all resources (77%, n = 43). The most frequently reported surgical treatment was burr holes (73%, n = 41). HCP resources were more likely to report the use of Twist Drill craniotomy compared to patient resources (66% vs 8%, p<0.001). There was no significant difference in reporting of rationale for surgical approach (39% vs 30%, p = 0.505), burr holes (62% vs 83%, p = 0.066),

Craniotomy (54% vs 63%, p = 0.472), craniectomy (8%, vs 7%, p = 0.882), and surgical drain use (15% vs 33%, p = 0.122). There was no reporting on anaesthetic use in either resource type.

**Domain 6: Nonsurgical management.** The most commonly reported sub-domains in nonsurgical management was conservative management, with no intervention (32%, n = 18). HCP resources had higher reporting of steroids and dexamethasone use (40% vs 12%, p = 0.016), Middle meningeal artery (MMA) embolization (37% vs 0%, p<0.001), and other medications (47% vs 8%, p = 0.001).

**Domain 7: Complications and recurrence.** The most frequently reported sub domains were complications- frequency and listing of complications (both 48%, n = 27). HCP resources were more likely to report all sub-domains, including frequency of recurrence (83% vs 42%, p<0.001), factors associated with recurrence (77% vs 12%, p<0.001), surgical treatment of recurrence (43% vs 4%, p<0.001), nonsurgical treatment of recurrence (40% vs 12%, p = 0.016), frequency of complications (63% vs 31%, p = 0.015), and list of complications (63% vs 31%, p = 0.015).

**Domain 8: Outcomes.** The most frequently reported outcomes were long term functional effects of a CSDH (50%, n = 28), and morbidity/quality of life (46%, n = 26). HCP resources reported information relating to survival more commonly than patient resources (53% vs 15%, p = 0.003). Patient resources provided information on support organisations more than HCP resources (46% vs 0%, p<0.001). There was no reporting on long-term anticoagulation management for either resource type.

## Discussion

In this scoping review we identified and analysed 56 educational resources, targeted at professionals, patients, carers and/or the public. These resources predominantly aim to focus on management strategies, and aetiology and epidemiology, with wide differences in coverage between HCP aimed resources, and patient resources. Information provision differed by and within audience type. HCP resources focused on surgical management and complications, whilst patient resources covered natural history, symptoms and diagnosis. Over eight in ten of resources provided information on the aetiology, epidemiology, and pathophysiology of CSDH; however less than half provided information on symptoms and diagnosis.

HCP resources focussed on management, surgical and/or non-surgical. This is in-line with the current focus of CSDH research; interventions or surgical techniques [26, 27] and adjuvant therapies such as steroids [28], antiepileptic medications [29] and statins [30]. Less than one-sixth of HCP resources included information on symptoms or diagnosis. Scientific articles are likely to be read by HCPs, particularly those with a specialist interest. Therefore, it is feasible that symptoms and diagnosis would be less commonly covered, due to inferred existing knowledge. However, this will not represent all HCPs that manage CSDH. Our findings elsewhere indicate the majority of patient HCP interactions in CSDH, are by HCP without a specialist interest in CSDH [31]. For example, most patients present with CSDH at non-specialist hospitals without access to a neurosurgical unit and are referred to a tertiary external neurosurgery unit (NSU) for advice, guidance, and potential transfer [10]. Not only do they often provide the direct and long-term care, but they will also facilitate shared decision making. In particular, literature is lacking on symptoms and diagnosis of CSDH- this may have an adverse impact on early recognition and diagnosis. Late diagnosis of CSDH has been associated with worse outcomes [32], and therefore recognising CSDH early, particularly in older patients with ambiguous presentations, is vital [33].

Assumptions on the needs of the audience were potentially present in the patient resources. Here information overwhelmingly focused on symptoms and diagnosis with little provided on

long term outcomes. This could be due to the intent of resources to inform patients and carers of the fundamental principles of CSDH, and not its long-term prognosis. However, this may not cover all information needs, for example when considering the decision of surgical management. Surgery has a reported recurrence rate of between 8 and 30% [13], with a risk of adverse effects and long term implications for cognition and quality of life [34, 35]. Such information would theoretically be important for effective shared decision making.

Within patient resources, further content potentially pertinent to shared decisions were missing. Examples included mode of anaesthesia, management of anticoagulation or antiplatelet use, but also non-surgical management strategies. This again could be due to the resource focus, but may also reflects the quality of evidence for these topics. However the availability of evidence is not necessarily a surrogate of need. UK national audit data suggests that although a majority of non-operative cases did not have surgery due to the small size of their CSDH, a significant portion did not have surgery due to either futility or best-interests decisions [11]. Arguably these groups have distinct information needs, and further work should explore what information individuals belonging to this non-operative cohort require. Frailty and baseline function highly influence this, and as they form part of this decision-making process, should also be considered in CSDH educational materials to facilitate intervention discussions among patients and families [36].

Middle Meningeal Artery (MMA) embolization and other experimental nonsurgical treatments are emerging management strategies for CSDH [37] and were not explored by patient resources, despite their increasing usage by some centres [37]. This would align with a preference for evidence or knowledge to inform educational content, rather than the clinical need [38].

In regard to reporting of symptoms, increasing age was the most reported risk factor in HCP resources. This has been demonstrated to be one of the most significant factors associated with CSDH development [39]. In contrast amongst patient resources, alcohol misuse was the most commonly reported risk factor. Whilst age may have been omitted as unmodifiable, the better-known associations of antiplatelets and anticoagulants was also rarely mentioned [40–42]. It is not known if patients are informed about these risks, which is crucial to promote shared decision making and patient autonomy.

Overall, the varied education provision in resources ultimately indicate an uncertainty over the exact educational need for CSDH amongst both patients and HCPs, with current provision emphasising CSDH as a whole disease process, as opposed to individual decisions required at points in the patient journey.

## Comparisons to previously published studies

These findings are in keeping with the broader, albeit limited neurosurgical literature. A previous systematic review examining surgeon-patient communication in neurosurgery identified that patient comprehension was low, and informational needs were often unmet, causing patient dissatisfaction [17]. This suggests that a mismatch of information can exist between clinicians and patients, despite information being available further raising questions around the preferred method of delivery. Limited studies of patient education resources have reported high satisfaction; but this still remains limited to self-produced resources [43]. Despite videos being rated by patients as the most effective medium for enforcing communication [44], there was only one resource available that utilised this approach. Patient based resources in CSDH have been scarcely researched, but studies examining Traumatic Brain Injury (TBI) based resources have demonstrated poor global quality, in keeping with the findings of our study [45].

## Limitations

Our study has several limitations; firstly, core domain selection was obtained by thematic analysis with the writing group, which did not include patients. This may result in skewed selection. However we tried to be as broad as possible in our definitions, and many of the domains identified and addressed within resources are highlighted as important in the James Lind Alliance priority setting partnership [46], indicating coverage of key patient priorities. Second, the search strategy was not all-encompassing, and does not include every published systematic review on CSDH, nor consider every digital health education resource. Physical resources such as patient self-help groups, and local patient care centres, or alternative health platforms would not also be identified. Whilst this may increase the probability of selection bias we feel this is a reflective sample sufficient to indicate the priorities and landscape of many CSDH articles.

Although this article provides evidence of the type of information provided and quantity of, it is still unclear as to how this information is received, and understood, by patients. This lack of evidence for translational efficacy is something that will be examined in due course [25]. Finally, one of two primary reviewers (CSG) had knowledge of CSDH- although this may have resulted in selective extraction and categorisation [47], dual extraction with a second reviewer not familiar with CSDH (SMc) did not identify any discrepancies.

## Perspective for future research

Overall, this work confirms the need to improve education resources to support CSDH clinical care. It highlights not only a probable need amongst patients, families and carers, but also non-specialist HCP, for clear and complete education resources. The identified eight content domains for CSDH education can be a foundation for such improvements, both as a framework to develop resources and articles moving forward, but also to research establishing their specific significance to different stakeholder groups, or decision points in CSDH care.

As outlined, shared decision making is critical to a condition like CSDH, where treatment decisions such as surgery need to be contextualised to many factors [15, 25, 35]. Enabling a patient or their significant others to be an equal participant in such a conversation can only occur once they have received sufficient education. Examples of effective tools that have been developed to support this include Core Information Sets, as well as patient decision aids [43, 48]. Implicit within their design is the assumption that the healthcare professional is the 'expert'. For CSDH, where shared decision making may be facilitated by non-specialists, this is not always the case, and this may need careful consideration during design and implementation. Addressing shared decision making for CSDH has been recognised as a major priority for improving care [49].

## Conclusions

This study highlights the available resources for patients, and healthcare professionals, and their content. Education resources for CSDH contain vary considerably, with clinician-focussed resources including surgical management; and natural history, and patient focussed resources on symptoms and diagnosis. These inconsistencies, aligned with the current clinical experience of educational deficiencies, indicate a need for improved content. The development of key information domains can support this on-going process.

## Supporting information

**S1 Checklist. Preferred Reporting Items for Systematic reviews and Meta-Analyses extension for Scoping Reviews (PRISMA-ScR) checklist.**
(DOCX)

**S1 Data.**
(XLSX)

# Acknowledgments

We would like to thank Dr Ali Alam, for providing feedback on the structure of the manuscript prior to submission.

# Author Contributions

**Conceptualization:** Samuel Khanna, Mark E. Vivian, Ellie Edlmann, Daniel J. Stubbs, Benjamin M. Davies.

**Data curation:** Conor S. Gillespie, Samuel Khanna, Mark E. Vivian, Samuel McKoy, Alvaro Yanez Touzet, Daniel J. Stubbs, Benjamin M. Davies.

**Formal analysis:** Conor S. Gillespie, Samuel Khanna, Mark E. Vivian, Samuel McKoy, Alvaro Yanez Touzet, Daniel J. Stubbs, Benjamin M. Davies.

**Funding acquisition:** Benjamin M. Davies.

**Investigation:** Samuel McKoy, Ellie Edlmann.

**Methodology:** Conor S. Gillespie, Benjamin M. Davies.

**Project administration:** Conor S. Gillespie, Mark E. Vivian, Alvaro Yanez Touzet, Benjamin M. Davies.

**Resources:** Conor S. Gillespie, Daniel J. Stubbs, Benjamin M. Davies.

**Software:** Mark E. Vivian, Alvaro Yanez Touzet, Daniel J. Stubbs.

**Supervision:** Conor S. Gillespie, Samuel Khanna, Mark E. Vivian, Alvaro Yanez Touzet, Ellie Edlmann, Daniel J. Stubbs, Benjamin M. Davies.

**Validation:** Conor S. Gillespie, Samuel Khanna, Alvaro Yanez Touzet, Daniel J. Stubbs.

**Visualization:** Conor S. Gillespie, Daniel J. Stubbs, Benjamin M. Davies.

**Writing – original draft:** Conor S. Gillespie, Samuel McKoy, Benjamin M. Davies.

**Writing – review & editing:** Conor S. Gillespie, Samuel Khanna, Mark E. Vivian, Alvaro Yanez Touzet, Ellie Edlmann, Daniel J. Stubbs, Benjamin M. Davies.

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
