## [Decision Letter · Decision Letter 0]

1 Dec 2022

PONE-D-22-18696Is information provided within chronic subdural haematoma education resources adequate? A scoping reviewPLOS ONE

Dear Dr. Davies,

Thank you for submitting your manuscript to PLOS ONE. After careful consideration, we feel that it has merit but does not fully meet PLOS ONE’s publication criteria as it currently stands. Therefore, we invite you to submit a revised version of the manuscript that addresses the points raised during the review process.

Both reviewers highlighted strengths of this manuscript. Yet, both of them, and especially reviewer 2, has made important recommendations for revisions that I kindly ask you to carefully address before I can make a decision about  publication. 

We look forward to receiving your revised manuscript.

Kind regards,

Sara Rubinelli

Academic Editor

PLOS ONE

Journal Requirements:

^2. We note that you have indicated that data from this study are available upon request. PLOS only allows data to be available upon request if there are legal or ethical restrictions on sharing data publicly. For more information on unacceptable data access restrictions, please see http://journals.plos.org/plosone/s/data-availability#loc-unacceptable-data-access-restrictions. ^

^In your revised cover letter, please address the following prompts:^

^a) If there are ethical or legal restrictions on sharing a de-identified data set, please explain them in detail (e.g., data contain potentially sensitive information, data are owned by a third-party organization, etc.) and who has imposed them (e.g., an ethics committee). Please also provide contact information for a data access committee, ethics committee, or other institutional body to which data requests may be sent.^

^b) If there are no restrictions, please upload the minimal anonymized data set necessary to replicate your study findings as either Supporting Information files or to a stable, public repository and provide us with the relevant URLs, DOIs, or accession numbers. For a list of acceptable repositories, please see http://journals.plos.org/plosone/s/data-availability#loc-recommended-repositories.^

^We will update your Data Availability statement on your behalf to reflect the information you provide.^

^3. ^Please include captions for your Supporting Information files at the end of your manuscript, and update any in-text citations to match accordingly. Please see our Supporting Information guidelines for more information: http://journals.plos.org/plosone/s/supporting-information.

Reviewers' comments:

Reviewer's Responses to Questions

**Comments to the Author**

1. Is the manuscript technically sound, and do the data support the conclusions?

Reviewer #1: Partly

Reviewer #2: Partly

2. Has the statistical analysis been performed appropriately and rigorously? 

Reviewer #1: Yes

Reviewer #2: Yes

3. Have the authors made all data underlying the findings in their manuscript fully available?

Reviewer #1: Yes

Reviewer #2: No

4. Is the manuscript presented in an intelligible fashion and written in standard English?

Reviewer #1: Yes

Reviewer #2: No

5. Review Comments to the Author

Reviewer #1: The manuscript addresses an important gap in knowledge in CSDH information resources, and has highlighted some important findings and implications for future research. The limitations of the study have been discussed fairly well. Some points for consideration are as follows-

1. Authors have stated that ‘Selected websites were commonly accessed educational tools based on existing knowledge and experience of authors’. This is likely to have created a bias in content search. How do the authors justify this?

2. It is mentioned in the text that ‘Videos and health education websites were identified by carrying out a search using Google (CA, USA) on 9th July 2021, including the top 3 results pages’ however, it is mentioned in the figure 1 that first 10 pages were searched.

3. An important data source of self-help groups and patient care centres seems to have been excluded from the search. These centres would potentially provide information to patients, their relatives and caregivers.

4. It should be clearly mentioned how many materials were searched to identify the 56 information sources discussed in this work.

5. It is mentioned under the sub-heading ‘Core domains’ that ‘Some were designed for patients, carers and the public (39%, n=22)’. However, elsewhere it is mentioned that there were 26 patient-targeted resources.

6. ‘Risk factors’ has been identified as a separate sub-domain under domain 2. The domain 2 also has separate sub-domains for alcohol use, head injury, hypotension etc. How is the sub-domain of ‘risk factors’ different from these.

7. The statement ‘For risk factors, patient resources described more alcohol misuse (62% vs 7%, p<0.001), and antiplatelet medication use (52% vs 17%, p=0.005) than patient resources’ needs correction.

8. It is not clear what is implied by the statement ‘The overall reporting across domains was 60.7%’.

Reviewer #2: This article deals with an interesting area, that certainly needs to be known in more depth for all the aspects described in the article (first and foremost, conscious future health care choices by the patient and family and early diagnosis by non-specialist healthcare professionals).

In general, the article has many aspects that could be improved.

Next time, I recommend the inclusion of lines in the manuscript, to facilitate the review.

I included review comments as an attachment.

Please see all comments in the attached file.

6. PLOS authors have the option to publish the peer review history of their article (what does this mean?). If published, this will include your full peer review and any attached files.

Reviewer #1: No

Reviewer #2: No

---

## [Author Response · Author response to Decision Letter 0]

31 Jan 2023

Specific Comments:

- Reviewer: 1

- The manuscript addresses an important gap in knowledge in CSDH information resources, and has highlighted some important findings and implications for future research. The limitations of the study have been discussed fairly well

Thank you very much for this comment- we are grateful for the comprehensive review of the manuscript. 

- 1. Authors have stated that ‘Selected websites were commonly accessed educational tools based on existing knowledge and experience of authors’. This is likely to have created a bias in content search. How do the authors justify this?

Thank you for this comment. The websites were selected as prominent patient facing, health education resources, based on popularity from internet search engines. These were identified as part of a similar review in Degenerative Cervical Myelopathy. We acknowledge however that this is not ‘comprehensive’, nor necessarily fully aligned with the CSDH population. We have expanded on this in the limitations section, and clarified this in the methods. 

- 2. It is mentioned in the text that ‘Videos and health education websites were identified by carrying out a search using Google (CA, USA) on 9th July 2021, including the top 3 results pages’ however, it is mentioned in the figure 1 that first 10 pages were searched.

Our apologises for this error. We have amended this to the first 10 pages, as originally intended.

- 3. An important data source of self-help groups and patient care centres seems to have been excluded from the search. These centres would potentially provide information to patients, their relatives and caregivers.

Thank you. This is possible, but difficult to quantify particularly for a relatively infrequency diagnosis. As our aim was to identify the most commonly encountered resources available, a focus on websites accessible via searching (which is recognised as the predominant source of health information today) is pragmatic but we believe robust. We have expanded on this in the limitations section of the manuscript, Line 372-374. 

- 4. It should be clearly mentioned how many materials were searched to identify the 56 information sources discussed in this work.

Our apologies- this has now been provided in the results section, Line 200.

- 5. It is mentioned under the sub-heading ‘Core domains’ that ‘Some were designed for patients, carers and the public (39%, n=22)’. However, elsewhere it is mentioned that there were 26 patient-targeted resources.

Our apologies for this. Yes, the correct number of patient-targeted resources was in fact 26, and not 22. We have amended this in the core domains sub heading (Line 213). 

- 6. ‘Risk factors’ has been identified as a separate sub-domain under domain 2. The domain 2 also has separate sub-domains for alcohol use, head injury, hypotension etc. How is the sub-domain of ‘risk factors’ different from these.

Thank you . Essentially the hierarchy is as follows: the individual risk factor (e.g. head injury) was coded. These can be collapsed into the sub-domain ‘risk factors’ (e.g. resource mentioned any risk factor at least once), and again into the domain ‘Natural History and Risk Factors’ (e.g. resources covering any theme within this domain). 

Whilst we acknowledge this is a little ambiguous, we feel descriptively the domain title ‘Natural History and Risk Factors’ more clearly articulates the content. 

- 7. The statement ‘For risk factors, patient resources described more alcohol misuse (62% vs 7%, p<0.001), and antiplatelet medication use (52% vs 17%, p=0.005) than patient resources’ needs correction.

Our apologies for this- this has now been corrected. 

- 8. It is not clear what is implied by the statement ‘The overall reporting across domains was 60.7%’.

Our apologies for the confusion- the authors meant that, when taking the overall % for each core domain covered by the resources (e.g 60% of resources cover natural history, 40% cover surgical management), the average of these was 60.7%. We have removed this statement due to its ambiguity from the revised version. 

Reviewer #2

- This article deals with an interesting area, that certainly needs to be known in more depth for all the aspects described in the article (first and foremost, conscious future health care choices by the patient and family and early diagnosis by non-specialist healthcare professionals).

Thank you for this. We agree that this article emphasises an under reported area in CSDH research. 

- Next time, I recommend the inclusion of lines in the manuscript, to facilitate the review.

The revised manuscript now included line numbers, and all changes have been marked with the line number they occur on for convenience. 

Abstracts 

- Background: it is said that CSDH is becoming prevalent, I would add a small sentence on why this is happening.

This has now been expanded upon (Lines 31-33 and 78-80). 

Objectives:

- I would remove the fact that 'they are a first step'. I would include ‘analysis and exploration’ (since you are also going to use qualitative methods and you also use an explorative approach). 

This statement has now been clarified to include that the aim of this paper is primarily analysis and exploration (Lines 38-40).

Introduction 

Third paragraph 

- (For CSDH this...), it is reported that neurosurgery is a third speciality, I would include more details on what this entails. 

Thank you for this- more details have been provided (Lines 92-95). 

- (These pose additional challenges in CSDH), I would add because it poses additional challenges. 

Thank you for this. On reflection this sentence is confusing, and we have modified completely. (Lines 99-100). 

Fourth paragraph 

- (However...) must acquire sufficient knowledge.... in order to... (I would add details/reasons). 

Thanks, however, we feel this is implied from the start of the sentence. 

The SDM process includes relative comparisons of treatments for example. What we are trying to articulate is that without the patient understanding the nature of their condition, this cannot take place. 

- (....decision-making process) I would add in '... among patients and families'. (In CSDH....unkown) I would replace unkown with a more appropriate term such as 'poorly researched', 'little information is available about it...', as something is available and exists anyway (even if not enough). 

These have now been added (Lines 108-109, and Line 139)

- (In this scoping...), I would change to '...we therefore explored the information in educational resources for CSDH'. 

This has now been added. Thank you (Line 110-111). 

- (To consider.... relatives), as the purpose of this study is not to research themes, but these are part of the results. I would remove this sentence and put it in the methods section. (The review therefore....) I would recommend rephrasing the specific objectives in a clearer and more direct way. I would also include a specification of when the objective concerns patients, health professionals and when it concerns both. 

This has been added to the methods section (Lines 124-125). Objectives have also been rephrased, and the target of each objective (patients, health professionals and both) has been added in parenthesis to each objective- (Lines 113-117). 

Materials and Methods 

- First paragraph: not clearly formulated. It is unclear what is meant by potential educational information and which and why it is said some publication to be defined as scientific publications (are they or are they not?). It is not clear from the paragraph what is included and what is not in the review. 

Our apologies for the confusion. We have since clarified this, that scientific publications are published articles in peer reviewed journals, and educational information in any other format. (Lines 123-124). 

- As specified the protocol has not been published, but can perhaps be included as supplementary information. 

Our apologies a formal study protocol in document form was not prepared for this study. 

Search strategy 

- the use of a search methodology is indicated, please provide scientific references supporting its use. 

Thanks we have added further clarification to this in the methods, and references. The approach is based on the experiences of others exploring information content in health conditions. 

Figure 1 

- It is not clear to the reader why the diagram is structured like this: did you start with the hospital searches that influenced subsequent searches? the diagram structured in this way could appear confusing, unless you justify why the structure is made this way (if this was done with a specific purpose). Otherwise, I would consider changing and structuring it more clearly. 

Apologies for this. We agree that the diagram structure looks confusing. The diagram is simply meant to show that all of these resources were searched independently, and the combined results amalgamated into a final list of resources. We have corrected this so it is structured more clearly. 

Resource type acquisition and searching 

- Only narrative reviews were included: I would include the rationale for this choice, which to me seems to take away much of the other results potentially available in the literature. 

We included narrative reviews as they were thought more likely to be broadly educational in nature, and therefore help establish if there was an information mis match between patients and professionals . This has been added to lines 128-130. 

Figure 2 

- It is not clear whether the words with a light grey background are readable (I printed out the figure in A4 but cannot read these words). Please evaluate the readability of this graphic. 

Our apologies for this. We have evaluated the readability of the Figure (Figure 2 being the spider diagram)- we cannot seem to detect this difference when viewed electronically. Nonetheless, we have increased the size of all words included to attempt to ameliorate this issue, and removed any grey backgrounds. 

Discussion 

- In general, there is not enough evidence discussed to make sense of the results obtained, links to existing literature and perspectives for future research. 

We appreciate the reviewer’s comment- substantial revisions have been made to the discussion- focussing on existing literature, and we have 2 separate paragraphs, one being ‘comparisons to previously published studies’ (Line 351) as per STROBE guidelines, and a section discussing future research ‘perspective for future research’. (Line 390). 

- What is the message this scientific article carries? - I recommend rewriting this section, so that these elements are sufficiently considered. MMA (full word, if term not previously encountered). 

We have added a message of the paper in the first few lines of the discussion (Lines 289-291). MMA has also now been expanded (Line 335). 

Limitations and future directions 

First paragraph 

- (Second, the search strategy....): I would give more details as to why this method was used. Also, I wondered about the criteria for this selection, e.g. were the most recent ones included? 

The reason why this method was used was to best attempt to replicate a member of the public, patient, or caregiver, when looking for resources on CSDH. This has been added to the paragraph (Lines 375). 

Second paragraph 

- (Third, although....) I would not include this as a limitation as this is not part of the objectives of this study and therefore not a limitation (but part of a future study perhaps). If anything, it is a paragraph that could be included in the section 'perspective for future research'. 

Thank you for this suggestion- we have added an additional subheading ‘Perspective for future research’ in accordance with the reviewers suggestion (Line 390). 

- (Finally,....) I would add reasons why the fact that one of the researchers knows the topic influences the results (on what assumption is this statement based and perhaps some references that support it). As said, if this limitation is confirmed and supported, references are needed. I would also include a sentence about how the authors tried to overcome and reduce the risk of this influence. 

Thank you for this- we have expanded on this. (Lines 386-388). 

- (This emcompass....analysis): I think this sentence is not needed here (this could be relevant to the Methods section). 

This has now been removed. Thank you. 

- (This review...process) I would state here that …in general future research perspectives in this area are.... 

This has now been added (Lines 401-402). 

- I would not include ‘a link’ to a specific future study, which has not yet been published and about which we do not know when it will be available in scientific literature. 

We agree with this viewpoint: this has now been modified to say that this could represent a future area of research, without linking the present article to a future study that has not yet happened yet. 

- I would add a few sentences on the value of the results in the clinical perspective and with respect to the use of services (I recommend to reflect on this and make a connection between the results and the aforementioned areas). 

This has been reflected on (Line 399). 

Conclusion 

- In my opinion, it should be reformulated. 

Thank you for this. We have reformatted the conclusion accordingly. 

- I would add what is the added value of this study and the message it carries, as well as the value of these results in a general way. 

This has been added to line 398. 

- I would consider the inclusion of 'Supplementary table S1. Differences in domain components between patient orientated and healthcare professional orientated resources.' in the article itself, as this is part of the focus of the article. If preferred as supplementary, argue why and make a clear link within the manuscript on this to invite the reader to go and see the attached table. 

Thanks- we have now included the Supplementary Table 1 as a table in the manuscript itself (Table 2) 

- All the recommended changes can make this article more readable and clear for the reader.

We hope the incorporation of these changes will make the manuscript more clear after the reviewer’s suggestions.

---

## [Decision Letter · Decision Letter 1]

7 Mar 2023

PONE-D-22-18696R1Is information provided within chronic subdural haematoma education resources adequate? A scoping reviewPLOS ONE

Dear Dr. Davies,

Thank you for submitting your manuscript to PLOS ONE. After careful consideration, we feel that it has merit but does not fully meet PLOS ONE’s publication criteria as it currently stands. Therefore, we invite you to submit a revised version of the manuscript that addresses the points raised during the review process.

The reviewers much appreciated the revisions. But one of them still has some suggestions that I kindly ask you to address before I can make the final decision on publication. 

We look forward to receiving your revised manuscript.

Kind regards,

Sara Rubinelli

Academic Editor

PLOS ONE

Journal Requirements:

Reviewers' comments:

Reviewer's Responses to Questions

**Comments to the Author**

1. If the authors have adequately addressed your comments raised in a previous round of review and you feel that this manuscript is now acceptable for publication, you may indicate that here to bypass the “Comments to the Author” section, enter your conflict of interest statement in the “Confidential to Editor” section, and submit your "Accept" recommendation.

Reviewer #1: All comments have been addressed

Reviewer #2: (No Response)

2. Is the manuscript technically sound, and do the data support the conclusions?

Reviewer #1: Yes

Reviewer #2: Yes

3. Has the statistical analysis been performed appropriately and rigorously? 

Reviewer #1: Yes

Reviewer #2: N/A

4. Have the authors made all data underlying the findings in their manuscript fully available?

Reviewer #1: Yes

Reviewer #2: No

5. Is the manuscript presented in an intelligible fashion and written in standard English?

Reviewer #1: Yes

Reviewer #2: Yes

6. Review Comments to the Author

Reviewer #1: (No Response)

Reviewer #2: An external file with the comments has been attached.

7. PLOS authors have the option to publish the peer review history of their article (what does this mean?). If published, this will include your full peer review and any attached files.

Reviewer #1: No

Reviewer #2: No

---

## [Decision Letter · Decision Letter 2]

21 Mar 2023

Is information provided within chronic subdural haematoma education resources adequate? A scoping review

PONE-D-22-18696R2

Dear Dr. Davies,

We’re pleased to inform you that your manuscript has been judged scientifically suitable for publication and will be formally accepted for publication once it meets all outstanding technical requirements.

Kind regards,

Sara Rubinelli

Academic Editor

PLOS ONE

Additional Editor Comments (optional):

Reviewers' comments:

Reviewer's Responses to Questions

**Comments to the Author**

1. If the authors have adequately addressed your comments raised in a previous round of review and you feel that this manuscript is now acceptable for publication, you may indicate that here to bypass the “Comments to the Author” section, enter your conflict of interest statement in the “Confidential to Editor” section, and submit your "Accept" recommendation.

Reviewer #1: All comments have been addressed

2. Is the manuscript technically sound, and do the data support the conclusions?

Reviewer #1: (No Response)

3. Has the statistical analysis been performed appropriately and rigorously? 

Reviewer #1: (No Response)

4. Have the authors made all data underlying the findings in their manuscript fully available?

Reviewer #1: (No Response)

5. Is the manuscript presented in an intelligible fashion and written in standard English?

Reviewer #1: (No Response)

6. Review Comments to the Author

Reviewer #1: (No Response)

7. PLOS authors have the option to publish the peer review history of their article (what does this mean?). If published, this will include your full peer review and any attached files.

Reviewer #1: No

---

## [Editor Report · Acceptance letter]

28 Mar 2023

PONE-D-22-18696R2 

Is information provided within chronic subdural haematoma education resources adequate? A scoping review 

Dear Dr. Davies:

I'm pleased to inform you that your manuscript has been deemed suitable for publication in PLOS ONE. Congratulations! Your manuscript is now with our production department. 

Kind regards, 

on behalf of

Dr. Sara Rubinelli 

Academic Editor

PLOS ONE